# Temporal and Spatial Differences in $CO_2$ Equivalent Emissions and Carbon Compensation Caused by Land Use Changes and Industrial Development in Hunan Province

Huangling Gu [1,2], Yan Liu [1], Hao Xia [1], Zilong Li [1], Liyuan Huang [1] and Yanjia Zeng [1,*]

1 School of City and Environment, Hunan University of Technology, Zhuzhou 412007, China; 14071@hut.edu.cn
2 School of Metallurgy and Environment, Central South University, Changsha 410083, China
* Correspondence: zengyanjia@hut.edu.cn; Tel.: +86-731-2218-3167

**Abstract:** The differences in $CO_2$ equivalent emissions and carbon compensation due to land use changes can provide a basis for formulating low-carbon development policies in various regions according to net $CO_2$ emissions. Based on the land use and energy consumption data of Hunan Province from 2000 to 2020, the calculation model is constructed to calculate the $CO_2$ equivalent emissions and carbon compensation values for different cities in different periods. The results showed that: (1) From 2000 to 2020, there was a significant growth trend in the area of built-up land in Hunan Province, mainly from the forest and cropland, while the area of forest is relatively stable. (2) The net $CO_2$ equivalent emissions from land use changes in Hunan Province shows a trend of increasing first and then decreasing with an initially fast and then slowed growth rate. Built-up land is the main carbon source, and its $CO_2$ equivalent emissions increased by 26.78 million tons, while the forest is the main carbon sink, and its carbon absorption decreased by $4.11 \times 10^4$ tons. (3) The carbon sink areas are mainly located in Zhangjiajie and Xiangxi in western Hunan, and the $CO_2$ equivalent emission intensity of other carbon source areas is gradually increasing from eastern Hunan to southern Hunan. (4) The carbon compensation value is very similar to the spatial distribution of $CO_2$ equivalent emissions in different cities. The high carbon compensation areas are mainly Yueyang, Loudi, and Xiangtan due to their backward energy structure, un-upgraded industrial structure, and large net $CO_2$ equivalent emissions, while the high carbon compensation areas are mainly Changsha, due to its high economic development level, great technological progress, and small net $CO_2$ equivalent emissions. To achieve regional coordination and low-carbon development, it is necessary to continually improve the carbon compensation mechanism and to build on carbon compensation to promote regional low-carbon coordinated development from a low-carbon level. Meanwhile, the government should rank and direct the transformation and development of different types of cities, to build a low-carbon land development model and achieve the goal of developing carbon neutrality.

**Keywords:** $CO_2$ equivalent emission; carbon source/sink; land use changes; carbon compensation; Hunan Province

## 1. Introduction

Climate change, particularly global warming, is a serious challenge for humanity and a global environmental problem recognized by the international community [1,2]. The large amount of $CO_2$ produced by fossil fuels and other energy sources consumed by human activities, although not directly reflected by $CO_2$ equivalent emissions through land use, is based on land use as the carrier and is the direct result of human activities [3,4]. In the context of global warming, it is important to build a low-carbon economic development model for cities. According to the Global Carbon Project (GCP), the $CO_2$ equivalent emitted by land use change is the major cause of global climate change and the increased greenhouse effect, while the research of World Resources Organization and experts related to the carbon cycle found that the $CO_2$ equivalent emissions caused by land use change

account for almost a third of the total greenhouse gas emissions, and second only to fossil fuel combustion [5–7]. The Fourteenth Five-Year Plan for the National Economy issued at the end of 2020, clearly stated that China is projected to achieve peak carbon emissions by 2030 and carbon neutrality by 2060. Currently, it is difficult to alter the fossil-based economic development model in the short term, and it has become a hotspot to study the overall changes in and drivers of land use emissions, to reduce $CO_2$ equivalent emissions by optimizing and adjusting the land use structure, and to build a national low-carbon space system to achieve sustainable socio-economic development.

The theme of the influence of land use change on $CO_2$ equivalent emissions has received wide attention from domestic and foreign researchers. The impact of $CO_2$ equivalent emissions resulting from regional land use changes was studied in the 1990s [8,9], and it was found that there were important differences in the impact of different land use patterns on $CO_2$ equivalent emissions from terrestrial ecosystems, and the expansion of built-up land had the most significant impact on regional $CO_2$ equivalent emissions. Campbell et al., (2000) [10] believed that net $CO_2$ equivalent emissions were mainly dependent on land use changes caused by human activities, including deforestation, cropland use, and cropland flowing into forests, followed by the continued growth in $CO_2$ concentration and other natural impact processes. Relevant domestic studies have studied the mechanism of land use emissions, $CO_2$ equivalent emission accounting, $CO_2$ equivalent emission effect, and $CO_2$ equivalent emission influencing factors at various scales, such as country, economic zone, and city [11–13]. On the national scale, Ge et al., (2008) [14] found that the changes in land and vegetation cover in China's terrestrial ecosystems during the past 300 years caused $CO_2$ equivalent emissions of 9.54 PgC, and the rapid expansion of reclamation and the rapid reduction in forests both had an impact on the terrestrial ecosystem carbon cycle after using LUCC (land use and land cover change) data reconstructed using historical documents in 2008. Provincially, the influence of land use change on $CO_2$ equivalent emissions has been assessed in the southwestern provinces of Sichuan and Chongqing [15], the eastern provinces of Jiangsu and Fujian [16], and the western provinces of Shaanxi and Shanxi [17]. However, the rapidly developing provinces in the central region have received less attention. Based on the findings, although there is a consensus on the profound impact of land use change on $CO_2$ equivalent emissions, significant differences exist in the spatial and temporal characteristics of $CO_2$ equivalent emissions in various regions and provinces; for example, the spatial features of $CO_2$ equivalent emissions in Jiangsu province show an increasing trend in the north compared to the south [16], while Sichuan province has the largest $CO_2$ equivalent emissions in the central region [15]. Liu et al. (2017) [18] and Zhou et al., (2018) [19] studied the land use emissions in Northeast China and Xi'an City by $CO_2$ equivalent emission coefficient method at the regional and city scales, respectively, and found that Northeast China and Xi'an City face a greater $CO_2$ equivalent emission pressure. In addition, due to differences in land use and levels of economic development in different provinces, the effects of $CO_2$ equivalent emissions in the region are also different, indicating that the formulating of emission reduction strategies based on the spatiotemporal characteristics of $CO_2$ equivalent emissions has obvious regional significance.

Current research on land use emissions mainly includes the spatial and temporal changes in the different effects of land use emissions [20], the characteristics and drivers of land use emissions [21], the connection between land use emissions and economic development [22], carbon footprint [23], etc. Research methods include the gray theory model [24], LMDI model [25], SBM model [26], panel data regression model [27], etc. However, in view of the problems of ecological environment and economic development, it is necessary to continue research into how to combine spatiotemporal differences in land use emissions with economic development, achieve carbon compensation, and promote the development of a low-carbon economy. As a typical demonstration province in central China, Hunan Province is also facing a rapid growth in net equivalent $CO_2$ emissions in the process of urbanization and industrialization. Therefore, this study analyzes the

land use changes in Hunan Province over the last 20 years from 2000 to 2020 from the perspective of dynamic land use change. At the same time, $CO_2$ equivalent emission and carbon compensation produced by industrial development and land use change in Hunan Province are estimated, so as to explore the spatiotemporal evolution characteristics of $CO_2$ equivalent emissions in Hunan Province under different land use patterns and to carry out carbon compensation zoning based on the law of $CO_2$ equivalent emissions on economic development, and then to discuss the adjustment and optimization of land use from a low-carbon economic perspective. This study can provide a scientific reference for Hunan Province's low-carbon goal-oriented spatial planning and control strategy, which is of great importance in promoting a low-carbon spatial development model and a low-carbon society development.

## 2. Materials and Methods

### 2.1. Overview of the Study Region

　　Hunan Province is situated in central China between $108°47'$–$114°15'$ E and $24°38'$–$30°08'$ N, in the middle reaches of the Yangtze River, which is bordered by Jiangxi Province to the east, Hubei Province to the north, Guizhou and Chongqing to the west, and Guangxi and Guangdong to the south (Figure 1). Hunan Province contains 14 cities and prefectures, namely Changsha City, Zhuzhou City, Xiangtan City, Hengyang City, Shaoyang City, Yueyang City, Chengde City, Zhangjiajie City, Yiyang City, Chenzhou City, Yongzhou City, Huaihua City, Loudi City, Xiangxi Tujiazu, and Miaozu Autonomous Prefecture. With an area of 211,800 km$^2$ in 2019, Hunan Province had 4.155 million hectares of cropland, accounting for about 3% of the total cropland in China, and 11.1236 million hectares of forest, accounting for about 5.1% of the total forest in China. It is very rich in plant species, with a complex flora and various kinds of vegetation, such as subalpine, mesic coniferous forests, deciduous broad-leaved forests, mixed coniferous and broad-leaved forests, and a considerable number of gully and valley evergreen broad-leaved forests and well-preserved scenic forests. The total $CO_2$ equivalent emissions amounted to $3.1 \times 10^8$ tons in 2019. The majority of Hunan Province is a terrestrial ecosystem with abundant natural resources, such as forests, wetlands, and grasslands, which play a significant role in carbon cycling and the reduction of carbon concentration. During the last 20 years, the resident population of Hunan Province increased from 65.62 million to over 66.45 million, and the total GDP surged from 355.149 billion yuan to 4154.257 billion yuan, while the urban built-up areas expanded significantly, with a high proportion of industrial output, and the service industry had become a new growth engineer and was in a critical period of transformation [28,29]. Currently, Hunan Province is going through a critical period of rapid urbanization, so it is of great value for research and of practical importance to study the spatial and temporal development pattern of carbon neutrality in the context of land use change.

### 2.2. Data Sources

　　This study gathered and pre-processed land use and carbon emissions data of 14 cities in Hunan Province. Land use data were provided by the Resource and Environmental Science and Data Center of the Chinese Academy of Sciences (http://www.resdc.cn, accessed on 30 August 2022), and have a resolution of 30 m × 30 m, produced by manual visual interpretation and supervised classification methods with an accuracy verification of 85.72% and a Kappa coefficient of 0.82, including 6 first-level land types, such as cropland and forest, and 25 s-level land categories, such as paddy fields and dryland (http://www.resdc.cn, accessed on 30 August 2022, the land use classification is described in the Supplementary Material). After reclassification, the six classes were cropland, forest, built-up land, grassland, water, and not used, and land use data in 2000, 2005, 2010, 2015, and 2020 were obtained. Carbon emissions are calculated by the carbon emission factor multiplied by the activity level. Carbon emission factors include fossil fuel burning, energy consumption, industrial production process, human living, cattle respiration, and soil respiration, which were obtained from the *Hunan Province Statistical Yearbook*, *China Energy*

*Statistical Yearbook*, *Hunan Energy Statistical Yearbook*, and Hunan Energy Development Report between 2000 and 2020.

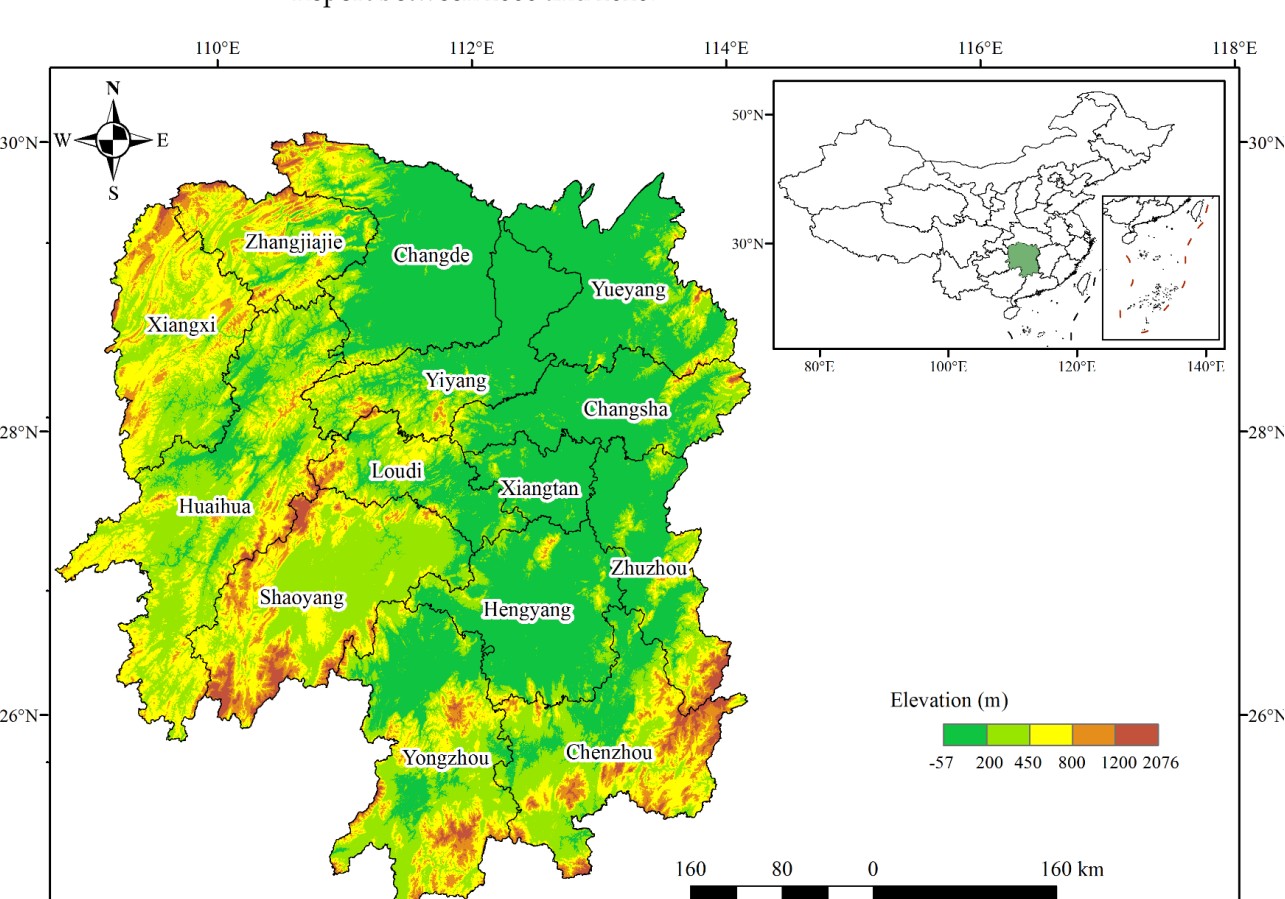

**Figure 1.** Location of Hunan Province.

### 2.3. Analysis Method

2.3.1. Land Use Change Rate

The single change rate of land use mainly relates to the quantitative change over a certain period or a certain land type, which can directly represent the speed of land use change, and its formula can be expressed as follows:

$$K = \frac{S_b - S_a}{S_a} * \frac{1}{T} * 100\% \tag{1}$$

where $K$ is the change rate of land use over a certain period; $T$ is the time interval within the study time, and the unit is a year; $S_a$ is the area of a land type at the initial stage of the study; $S_b$ is the area of a land type at the end of the study. A positive value of $K$ indicates an increase in land area over the study period, and a negative value of $K$ indicates a decrease in land area over the study period.

2.3.2. Calculation Method of $CO_2$ Equivalent Emissions

$CO_2$ equivalent emissions from land use change are closely related to cropland, forest, grassland, and built-up land. The accounting methods can be divided into the direct method and indirect method. The direct method refers to the emissions generated by the flow and maintenance of land use types (cropland, forest, grassland, water, not used), while the indirect method refers to the emissions generated by human activities on the built-up land area.

(1) Direct calculation method of $CO_2$ equivalent emission

For five types of cropland, forest, grassland, water and not used, the $CO_2$ equivalent emission formula is as follows:

$$C = S_I * E_I \tag{2}$$

where $C$ is the total $CO_2$ equivalent emission; $S_I$ is the area of each type of land use; $E_I$ is the $CO_2$ equivalent emission coefficient of each type of land use. According to previous research results [30–41], the $CO_2$ equivalent emission coefficient of cropland, forest, grassland, water, and not used is summarized (Table 1), for which the $CO_2$ equivalent emission coefficient is the quantity of carbon dioxide released per unit of carbon source factor, the latter usually measured in tons. The $CO_2$ equivalent emission coefficient of a carbon source is positive, while the $CO_2$ equivalent emission coefficient of a carbon sink is negative. The $CO_2$ equivalent emission coefficients in this study are the average of several literature data.

**Table 1.** $CO_2$ equivalent emission coefficients and documentary sources for each type of land use.

| Land Use Type | $CO_2$ Equivalent Emission Coefficients ($t \cdot hm^{-2}\ year^{-2}$) | Reference |
|---|---|---|
| Cropland | 0.497 | [30–32] |
| Forest | −0.587 | [33,34] |
| Grassland | −0.021 | [35,36] |
| Water | −0.253 | [37,38] |
| Not used | −0.005 | [36,39–41] |

(2) Indirect calculation method of $CO_2$ equivalent emission

The $CO_2$ equivalent emissions from built-up land are often estimated indirectly by the $CO_2$ equivalent emission coefficient of energy consumption in the process of utilization. According to the $CO_2$ equivalent emission estimation formula in the 2006 National Greenhouse Gas Emission Guidelines, the $CO_2$ equivalent emission accounting method for construction land can be obtained [42,43], and the formula is as follows:

$$C_e = \sum_{i=1}^{N} B_i * D_i * E_i \tag{3}$$

where $C_e$ is the total $CO_2$ equivalent emission of built-up land; $B_i$ is the energy usage; $D_i$ is the conversion ratio of the energy coal; $E_i$ is the $CO_2$ equivalent emission coefficient of the energy. Taking into account the real state of energy consumption in Hunan Province, in order to decrease the negative influence of statistical error, the energy types should be as completely chosen as possible, including 18 kinds of energy needed for production and life, such as raw coal, cleaned coal, other washed coal, coke, other coking products, crude oil, gasoline, coke oven gas, blast furnace gas, other gas, liquefied petroleum gas, kerosene, diesel oil, fuel oil, other petroleum products, natural gas, heat, and electricity. Meanwhile, the $CO_2$ equivalent emission coefficients of various energy sources are, respectively, referred to in the Guidelines for National Greenhouse Gas Emission Inventory issued by the IPCC in 2006 (The $CO_2$ equivalent emission coefficients of various energy are described in the Supplementary Material) [42,44], and the coal conversion coefficient of standard coal is referred to in the *China Energy Statistical Yearbook* [29].

### 2.3.3. Carbon Compensation Model

Net $CO_2$ equivalent emissions are taken as the basis for determining the carbon compensation benchmark [45]. If the net $CO_2$ equivalent emission is negative, this means that the carbon sink capacity of this region exceeds the carbon source, where the ecosystem can not only absorb the carbon emissions of its own region, but also absorb the carbon emissions of nearby areas. Therefore, this region should receive carbon compensation

funds; otherwise, it should pay carbon compensation funds [46]. The calculation formula is as follows:

$$L_i = E_{ci} - S_{ci} \tag{4}$$

where $L_i$ is the benchmark value of carbon compensation in region $i$; $E_{ci}$ is the carbon emission in region $i$ (t); Sci is the carbon removal in region $i$. When $L_i > 0$, carbon compensation funds should be paid; when $L_i < 0$, carbon compensation funds should be obtained.

However, due to the differences in economic development level, $CO_2$ equivalent emission intensity and $CO_2$ equivalent emission efficiency, only considering the net $CO_2$ equivalent emission as the benchmark value of carbon compensation, may cause excessive funds to be paid in some areas, which may lead to a certain deviation in the calculation results. Therefore, it is necessary to revise the benchmark value $L_i$ and set a $CO_2$ equivalent emission threshold $P_i$ [47] for each area, and the calculation formula is as follows:

$$P_i = ECC \times D = \frac{G_i}{G} \Big/ \frac{C_i}{C} \times D \tag{5}$$

where $P_i$ is the $CO_2$ equivalent emission threshold of each city (t); $ECC$ is the economic contribution coefficient of $CO_2$ equivalent emissions [47]; $D$ is the average $CO_2$ equivalent emission of all cities (prefectures) in the province (t); $G_i$ and $G$ are the GDP of each city (prefecture) and the GDP of Hunan Province (10,000 yuan), respectively; $C_i$ and $C$ are, respectively, the $CO_2$ equivalent emissions of each city and the total $CO_2$ equivalent emissions of Hunan Province (t).

Since there are spatiotemporal differences in $CO_2$ equivalent emission intensity between different cities, the $CO_2$ equivalent emission is revised according to the $CO_2$ equivalent emission intensity of different cities (prefectures) in 2015 and 2020 and its comparison with the $CO_2$ equivalent emission intensity of the whole province [46], and the calculation formula is as follows:

$$E_{ci}^1 = E_{ci} \times (G_{t1-i}/G_{t2-i} - G_{T1}/G_{T2} + 1) \times G_{t1-i}/G_T \tag{6}$$

where $E_{ci}^1$ is the corrected $CO_2$ equivalent emission of region $i$ (t); $G_{t1-i}$ and $G_{t2-i}$ are the $CO_2$ equivalent emissions per unit GDP of region $i$ in 2020 and 2015, respectively (t/10,000 yuan); $G_{T1}$ and $G_{T2}$ are the $CO_2$ equivalent emissions per unit GDP of Hunan Province in 2020 and 2015, respectively (t/10,000 yuan); $G_T$ is the average $CO_2$ equivalent emission per unit GDP of all cities in Hunan Province in 2020 (t/10,000 yuan). The revised benchmark value of carbon compensation is as follows:

$$L_i^1 = E_{ci}^1 - S_{ci} - P_i \tag{7}$$

If $L_i^1 > 0$, the local government should pay carbon compensation funds; if $L_i^1 = 0$, there is no need to pay, and no compensation fund can be obtained. If $L_i^1 < 0$, carbon compensation funds can be obtained.

According to research by Zhao et al., (2016) [46] and Yu et al., (2012) [48], the model is properly modified to obtain the calculation method of carbon compensation value, as follows:

$$M_i = |L_i^1| \times \alpha \times \gamma \tag{8}$$

where $M_i$ is the carbon compensation fund obtained or paid by the $i$-th region (10,000 yuan); $\alpha$ is the price per ton of carbon (yuan/t); $\gamma$ is the ecological compensation coefficient. Here, $\alpha$ is calculated as follows:

$$\alpha = \frac{(P_{max} + P_{min})}{2} \times \frac{G_{P1}}{G_{P2}} \tag{9}$$

where $P_{max}$ and $P_{min}$ are the maximum and minimum domestic carbon sink prices, respectively [49] ($10^4$ yuan/$10^4$ tons); $G_{P1}$ is the GDP per capita of Hunan Province in 2020 (10,000 yuan/person); $G_{P2}$ is the national GDP per capita in 2020 (10,000 yuan/person).

$\gamma$ is represented by the improved Pearl growth curve model [50,51], as follows:

$$\gamma_i = \frac{A_i}{(1 + e^{-t})} \tag{10}$$

where $\gamma_i$ is the carbon compensation coefficient of the $i$-th region; $A_i$ is the carbon compensation capacity, which is equal to the ratio of regional GDP to the GDP of Hunan Province; $e$ is the basis of natural logarithm; $t$ is the Engel coefficient of Hunan Province in 2020.

## 3. Results and Analysis

### 3.1. Spatialtemporal Analysis of Land Use Change in Hunan Province

3.1.1. Characteristics of Land Use Change

According to the statistical data of 6 first-level land types of 14 cities (prefectures) in Hunan Province, the land use situations of each city (prefecture) in Hunan Province are obtained (Table 2, Figure 2). By the end of 2020, the total land area of Hunan Province was 21,175 km$^2$, and the land use types were mainly forest, cropland, water, and grassland, among which forest accounted for the largest proportion, about 62.16%, mainly distributed in the green ecological barrier areas, such as the Nanling Moutain Range, Luoxiao Moutain Range, and Wuling Mountain Range [52], distributed in Huaihua, Chenzhou, Xiangxi, and Zhuzhou, while the forest land area of four cities and prefectures accounts for about 40.52% of the total area. The next largest proportions were cropland and grassland, accounting for 28.03% and 3.23% of the total land area, respectively. Cropland was mainly spread over the Dongting Lake Plain in northern Hunan and the hilly basin in central Hunan, while grassland was mainly distributed in Xiangxi, Zhangjiajie, Yongzhou, and Huaihua. The water area, built-up land, and not used accounted for 3.41%, 2.71%, and 0.46%, respectively. The water area was mainly distributed in Yueyang, Yiyang, and Changde in the Dongting Lake Basin, and built-up land was mainly distributed in the Changsha–Zhuzhou–Xiangtan urban agglomeration, while not used was mainly distributed in Yiyang, Yueyang, and Changde.

**Table 2.** Statistical result of land use change in Hunan Province (unit: km$^2$).

| Year | 2000 | 2005 | 2010 | 2015 | 2020 | Change Rate of Land Use from 2000 to 2020 |
|---|---|---|---|---|---|---|
| Cropland | 61,262 | 60,931 | 60,732 | 59,918 | 59,371 | −0.62% |
| Forest | 132,203 | 132,103 | 131,989 | 131,721 | 131,647 | −0.08% |
| Grassland | 7566 | 7534 | 7489 | 7389 | 6837 | −1.93% |
| Water | 7055 | 7242 | 7258 | 7324 | 7216 | 0.46% |
| Not used | 742 | 717 | 781 | 743 | 978 | 6.36% |
| Built-up land | 2799 | 3100 | 3378 | 4532 | 5746 | 21.06% |

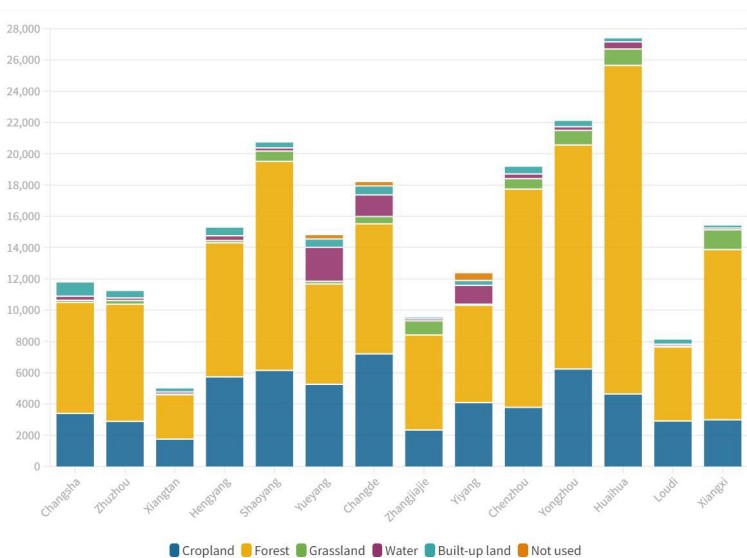

**Figure 2.** Each type of land use area of each city (prefecture) in Hunan Province in 2020.

During the last 20 years, the land use structure of Hunan Province has been constantly changing, and the land use change shows that the area of forest, cropland, and grassland decreased, while the area of the other three types of land use increased to varying degrees (Figure 3). The change in cropland showed a gradually decreasing trend, which decreased from 61,262 km$^2$ in 2000 to 60,732 km$^2$ in 2010, and further decreased to 59,371 km$^2$ in 2020, with a decrease of 1531 km$^2$ and a change rate of $-0.62\%$ in the last 20 years. The reduced areas were mainly distributed in Changde and Changsha, and years of cropland area reduction were mostly before 2010 (Figure 4). The grassland area decreased the most, from 7566 km$^2$ in 2000 to 6837 km$^2$ in 2020, with a reduction of 729 km$^2$ and a change rate of $-1.93\%$. The decreased areas were mainly Chenzhou and Huaihua. In contrast, the forest area decreased slightly, from 132,203 km$^2$ in 2000 to 13,1647 km$^2$ in 2020, with a decrease of 556 km$^2$ and a change rate of $-0.08\%$, and the decreased areas were mainly Chenzhou and Huaihua (Figure 4). The built-up land had shown a gradual increase trend, from 2799 km$^2$ in 2000 to 3378 km$^2$ in 2010, and then to 5746 km$^2$ in 2020, which doubled in the last 20 years, with an obvious growth rate of 21.06%. The urban expansion areas were mainly concentrated in Changsha, Zhuzhou, and Yueyang, and other Changsha–Zhuzhou–Xiangtan urban agglomerations (Figure 4).

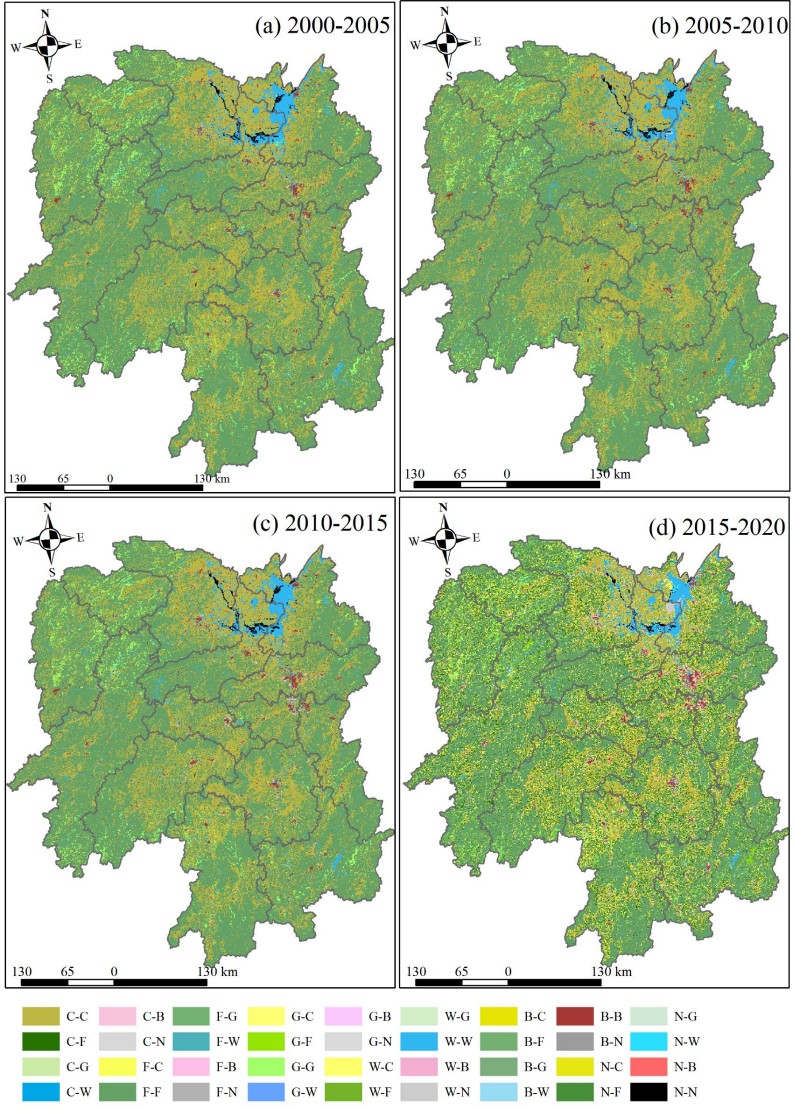

**Figure 3.** Spatial distribution of land use change in Hunan Province from 2000 to 2020. C—cropland, F—forest, G—grassland, W—water, B—built-up land, N—not used.

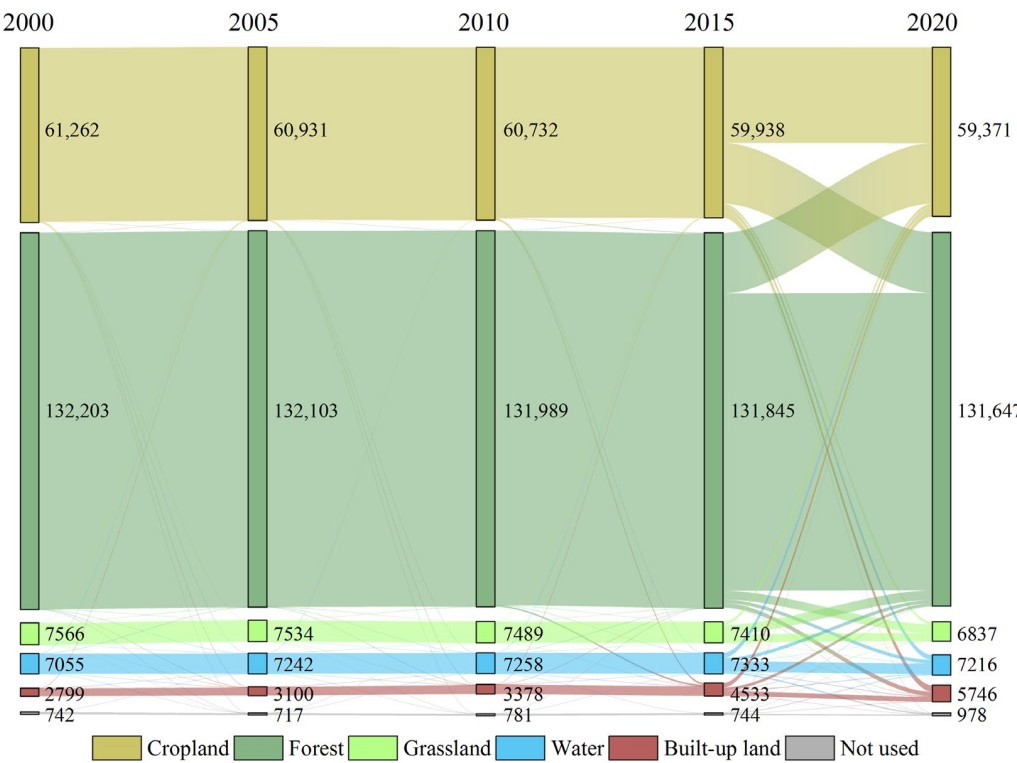

**Figure 4.** Sankey diagram of land use change in Hunan Province from 2000 to 2020 (unit: km$^2$).

### 3.1.2. Spatial Characteristics of Land Use Transformation

By analyzing the land use transfer matrix in Hunan Province, the transfer direction and transfer quantity of six major land use types in the study area can be obtained, as well as the flow of the transfer area of each land use transferred out and in (Table 3). During the last 20 years, the six land use patterns in Hunan Province have evolved, among which forest was the largest area of land use transferred out of the study area, followed by cropland, and the order of the number of transferred out areas was: forest > cropland > grassland > water > built-up land > not used. In the last 20 years, the largest area of land use transferred in of the study area was forest, followed by cropland, and the order of the number of transferred in areas was: forest > cropland > built-up land > grassland > water > not used.

**Table 3.** Transfer matrix of land use in Hunan Province during 2000–2020 (unit: km$^2$).

| | **Cropland** | **Forest** | **Grassland** | **Water** | **Not Used** | **Built-Up Land** | **Transfer Rate** |
|---|---|---|---|---|---|---|---|
| Cropland | *34,041* | 21,683 | 859 | 1893 | 71 | 2678 | 44% |
| Forest | 21,300 | *104,410* | 3050 | 1213 | 15 | 1851 | 21% |
| Grassland | 1035 | 3465 | *2819* | 112 | 4 | 58 | 62% |
| Water | 1695 | 1127 | 43 | *3511* | 450 | 204 | 50% |
| Not used | 43 | 7 | 1 | 259 | *427* | 8 | 42% |
| Built-up land | 1164 | 506 | 20 | 161 | 4 | *945* | 66% |

Note: The rows represent the amount of land areas converted from 2000 to 2020 in the land type units, and the columns represent the parts converted from 2000 to 2020 in the land type units. The slash & bold fonts represent the unchanged part.

From the flow of land use quantity, the cropland, forest, and grassland in Hunan Province decreased to varying degrees from 2000 to 2020, while the built-up land, water and, not used areas had an increasing trend. During the study period, cropland mainly flowed into forest, forest mainly flowed into cropland, and cropland and forest mainly flowed into built-up land in Hunan Province (Figure 4). Among them, 44.40% of cropland was transferred to other land use types. The higher transferred area was forest with an area of 21,683 km$^2$, followed by built-up land with an area of 2678 km$^2$, indicating that

with the growth in population and economic development, the demand for built-up land and other infrastructure land in Hunan Province is on the rise, and the influence of the policy of returning cropland to forest has led to a large area of cropland being flowed into forest; 20.80% of forest was transferred to other land uses, and the highest areas were cropland and grassland, with the transferred area reaching 21,300 km$^2$ and 3050 km$^2$, respectively. The land use types flowing into the construction land were mainly cropland and forest, with inflow areas of 2678 km$^2$ and 1851 km$^2$, respectively. During the study period, the increased areas of water mainly included Yueyang, Yiyang, Shaoyang, Loudi, and Chenzhou, and most of them flowed from cropland to a water area. On the one hand, this is due to the influence of policies, such as the return of cropland to lakes; on the other hand, people pay more attention to the protection of water resources and the quality of the ecological environment, resulting in a small increase in water area. The supplementary cropland mainly came from forest and water area, and a small part came from built-up land, showing that the additional cropland is mainly obtained through land development and consolidation, while the consolidation and reclamation of construction land is not deep enough. There was a small increase in not used land, mainly from water areas, which had been transferred to 450 km$^2$. The largest increase in not used land was mainly wetlands. Most of the tidal low-lying areas of wetlands were mainly distributed in Changde, Yueyang, and Yiyang near Dongting Lake. From 2000 to 2020, the wetlands in Yuanjiang County of Yiyang City, increased obviously with a wide range of distribution. However, the flowing of the water area into not used land was mainly due to the rewetting of water areas and the formation of herbaceous wetlands, resulting in a small increase in not used land.

### 3.2. Spatiotemporal Characteristics of $CO_2$ Equivalent Emissions in Hunan Province

3.2.1. Sequential Changes in $CO_2$ Equivalent Emissions in Hunan Province

Based on the direct and indirect method for calculating $CO_2$ equivalent emissions, the land use emissions in 14 cities (prefectures) of Hunan Province, as well as the total carbon sources, total carbon sinks, and net $CO_2$ equivalent emissions, are calculated, respectively (Table 4 and Figure 5). As can be seen from Table 4 and Figure 5a, the net $CO_2$ equivalent emissions of Hunan Province during the study period were affected by both carbon sources and carbon sinks. Among them, carbon sources showed a downward trend after increasing from $2728.79 \times 10^4$ tons in 2000 to $6620.97 \times 10^4$ tons in 2010, and then decreasing to $5396.97 \times 10^4$ tons in 2020. However, the carbon sinks showed a downward trend, from $-787.57 \times 10^4$ tons in 2000 to $-786.96 \times 10^4$ tons in 2010, and then to $-784.61 \times 10^4$ tons in 2020, with a small fluctuation range. In general, the carbon sources in Hunan Province were far greater than the carbon sinks, so the net $CO_2$ equivalent emissions of Hunan Province fluctuated from $1941.22 \times 10^4$ tons in 2000 to $4612.36 \times 10^4$ tons in 2020, which still resulted in a carbon source state.

**Table 4.** Changes in land use-based total annual $CO_2$ equivalent emissions in Hunan Province from 2000 to 2020 (unit: $10^4$ tons).

| Year | Forest Emissions | Grassland Emissions | Water Emissions | Not Used Emissions | Built-up Land Emissions | Cropland Emissions | Carbon Removal | $CO_2$ Equivalent Emissions | Net $CO_2$ Equivalent Emissions |
|------|------|------|------|------|------|------|------|------|------|
| 2000 | −768.10 | −1.59 | −17.85 | −0.04 | 2424.32 | 304.47 | −787.57 | 2728.79 | 1941.22 |
| 2005 | −768.39 | −1.58 | −18.32 | −0.04 | 4342.46 | 302.83 | −788.33 | 4645.29 | 3856.96 |
| 2010 | −766.86 | −1.58 | −18.36 | −0.17 | 6319.13 | 301.84 | −786.96 | 6620.97 | 5834.00 |
| 2015 | −766.02 | −1.56 | −18.55 | −0.04 | 5982.19 | 297.90 | −786.16 | 6280.08 | 5493.92 |
| 2020 | −764.87 | −1.44 | −18.26 | −0.05 | 5101.89 | 295.07 | −784.61 | 5396.97 | 4612.36 |

As shown in Figure 5b, net $CO_2$ equivalent emissions have been closely linked to carbon sources and carbon sinks of various types of land use. Among them, forest areas had the largest carbon sink effect, and accounted for over 95% of all carbon sinks. The total carbon sink of forest has shown a slow downward trend with a small fluctuation range, basically maintaining between $768.39 \times 10^4$ tons and $764.87 \times 10^4$ tons. Compared with

carbon sinks, the largest effect of carbon sources from was built-up land and cropland in turn. Among them, the carbon source effect of cropland showed a decreasing trend, from about 11.16% of the total $CO_2$ equivalent emissions in 2000 to 4.56% in 2010, and then to 5.46% in 2020. However, the carbon source effect of built-up land peaked in 2010 and then decreased, which grew from $2424.32 \times 10^4$ tons in 2000 to $6319.13 \times 10^4$ tons in 2010 and decreased to $5101.89 \times 10^4$ tons in 2020. Built-up land accounted for 88.83~95.44% of the total $CO_2$ equivalent emissions in Hunan Province, which indicates that the carbon emissions from built-up land play a decisive role in the change in net $CO_2$ equivalent emissions in Hunan Province.

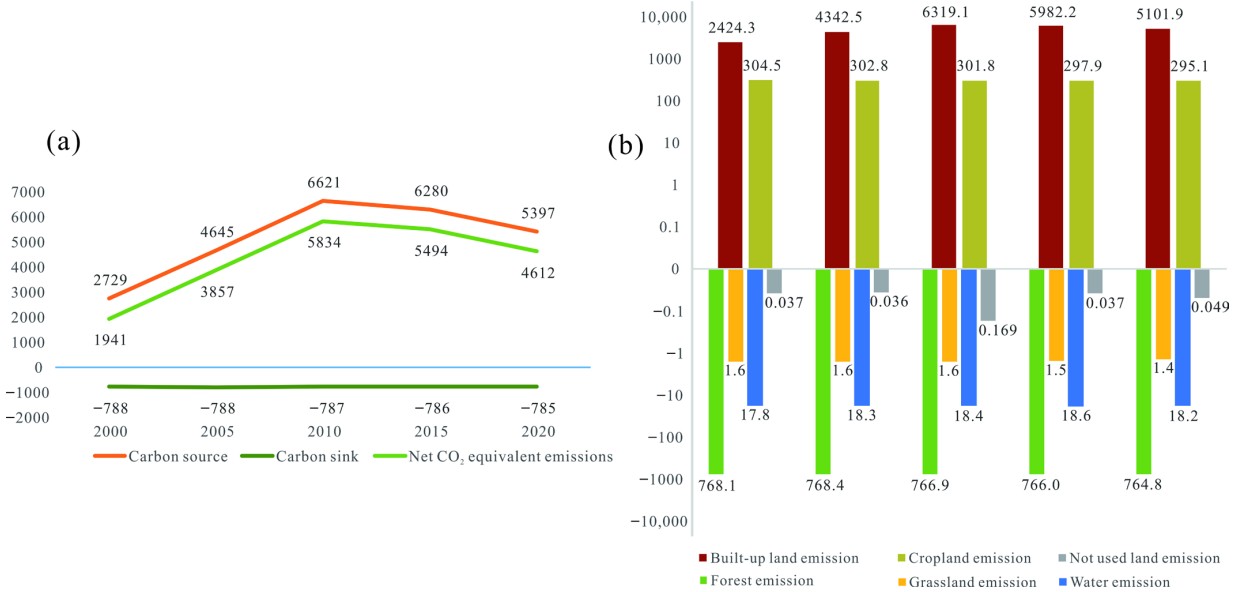

**Figure 5.** (**a**) Carbon source, carbon sink and net $CO_2$ equivalent emission in Hunan Province from 2000 to 2020 (unit: $10^4$ tons); (**b**) carbon emissions related to land use in Hunan Province from 2000 to 2020 (unit: $10^4$ tons).

3.2.2. Spatial Characteristics of $CO_2$ Equivalent Emissions in Hunan Province from 2000 to 2020

The land use emissions in Hunan Province showed a significant spatial heterogeneity from 2000 to 2020. Taking 2020 as an example (Figure 6), among the 14 cities (prefectures) in Hunan Province, Huaihua, Xiangxi and Zhangjiajie were carbon sink areas, with a wide range of forest distribution and relatively high carbon sequestration capacity. The other 11 cities were carbon source areas, which were mainly distributed in Yueyang, Loudi and other old industrial cities where traditional energy was the main source and the industrial structure was not optimized. The main carbon source was energy consumption on built-up land.

The $CO_2$ equivalent emissions in Hunan Province were usually high in the east and low in the west. The cities that had the greatest net $CO_2$ equivalent emissions were Yueyang and Loudi, with $1210.3 \times 10^4$ tons and $1136.9 \times 10^4$ tons, respectively, both exceeding 10 million tons, followed by Xiangtan, with a net $CO_2$ equivalent emission of $736.8 \times 10^4$ tons, while Yongzhou was the city with less net $CO_2$ emissions, with only $42.8 \times 10^4$ tons, while Huaihua, Xiangxi, and Zhangjiajie showed a carbon sink state as a whole, with net $CO_2$ equivalent emissions of $-46.1 \times 10^4$ tons, $-29.4 \times 10^4$ tons, and $-11.8 \times 10^4$ tons, respectively. The land use types of Yueyang and Loudi are mainly forest and cropland, and the built-up land shows an increasing trend, while the land use emissions increased by five and seven times from 2000 to 2020, respectively. From 2000 to 2020, Yueyang had the largest amount of forest flowed out, with a total area of 108 km², mainly flowed into cropland and built-up land; the second was cropland, with a flowed area of 69 km², mainly flowed into forest, water, and built-up land. The area of water

flowed out was 57 km², which was flowed into cropland and not used land. The flowed out area of grassland was relatively smaller. Among the flowed area, built-up land had the largest flowed area of 245 km², with 46.5% coming from cropland and 13.5% from forest, mainly concentrated in the surrounding areas of the urban built-up areas. Research had shown that with the rapid urbanization, Yueyang City mainly engages in economic construction by occupying cropland and forest, which will face an increasing pressure on carbon emissions reduction. Moreover, as Yueyang conducted economic development by resuming the industrial transfer of the city clusters in the middle reaches of Yangtze River, the large proportion of coal energy consumption and the lack of optimization of the energy structure have resulted in a high intensity of carbon emission and a low carbon sink effect, making it a high ecological pressure area. From 2000 to 2020, Loudi City had the largest area of cropland flowed out, with a total area of 99 km², mainly flowed into forest and built-up land; the second was forest, which was mainly flowed into cropland and built-up land; the flow area of built-up land was the largest. Loudi, as a resource-based city and the province's energy and raw material base, was an old industrial base. Therefore, there were more energy-intensive consuming industries, resulting in high carbon emissions.

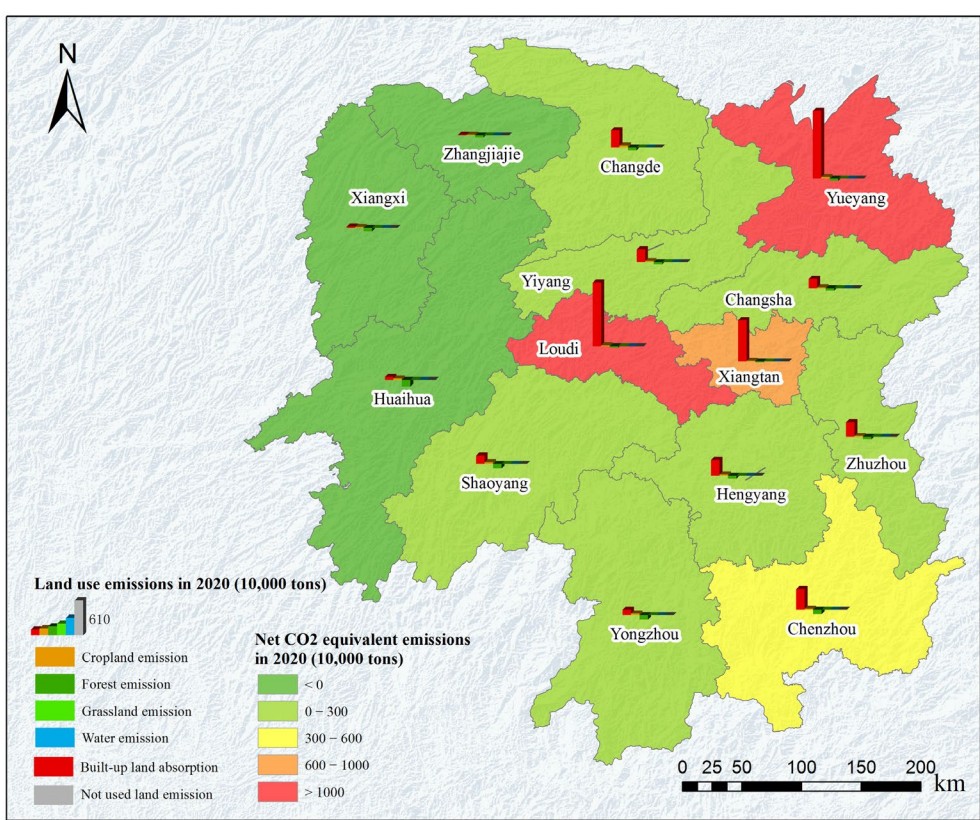

**Figure 6.** Spatial pattern and land use emission in Hunan Province in 2020.

From the histogram composition of net $CO_2$ equivalent emissions in various cities (prefectures), the carbon emissions were mainly caused by energy consumption on built-up land, and the carbon emissions in the two regions are far higher than the carbon sink, so the overall carbon source status was presented; The carbon sink of the three cities (prefectures) in the west of Hunan Province, namely Huaihua, Xiangxi, and Zhangjiajie, mainly came from the carbon sequestration of widely distributed forest. Due to the relatively weak industrial base in western Hunan Province, the carbon emission was much lower than the carbon sink, forming the spatial pattern of regional carbon sinks among 14 cities (prefectures) in Hunan Province.

In accordance with the spatial distribution scheme of carbon sources and sinks in various cities (prefectures) (Figure 7), the changes in $CO_2$ equivalent emissions in various

cities of Hunan Province showed significant differences from 2000 to 2020. During the study period, the $CO_2$ equivalent emissions in Yueyang continued to increase, rapidly increasing from $269.5 \times 10^4$ tons in 2000 to $1253.2 \times 10^4$ tons in 2020, with an obvious growth trend. Changsha, Zhuzhou, Xiangtan, Changde, Hengyang, Shaoyang, Yiyang, Chenzhou, Loudi, and Yongzhou showed an upward and then downward trend, all of which peaked in $CO_2$ equivalent emissions over the study period from 2010 to 2015. However, the $CO_2$ equivalent emissions in Zhangjiajie, Xiangxi, and Huaihua were decreasing. From 2000 to 2020, the carbon sink capacity of various cities in Hunan Province was relatively stable, among which the carbon sequestration capacity was high in western and southern Hunan, and low in eastern and central Hunan. Huaihua had the largest carbon sink, with over 1.22 million tons of $CO_2$ a year, while Loudi and Xiangtan had the smallest carbon sinks. According to the analysis of cities, the carbon absorption of Changsha, Chenzhou, Loudi, and Zhangjiajie decreased, while the other 10 cities maintained a stable carbon sequestration pattern.

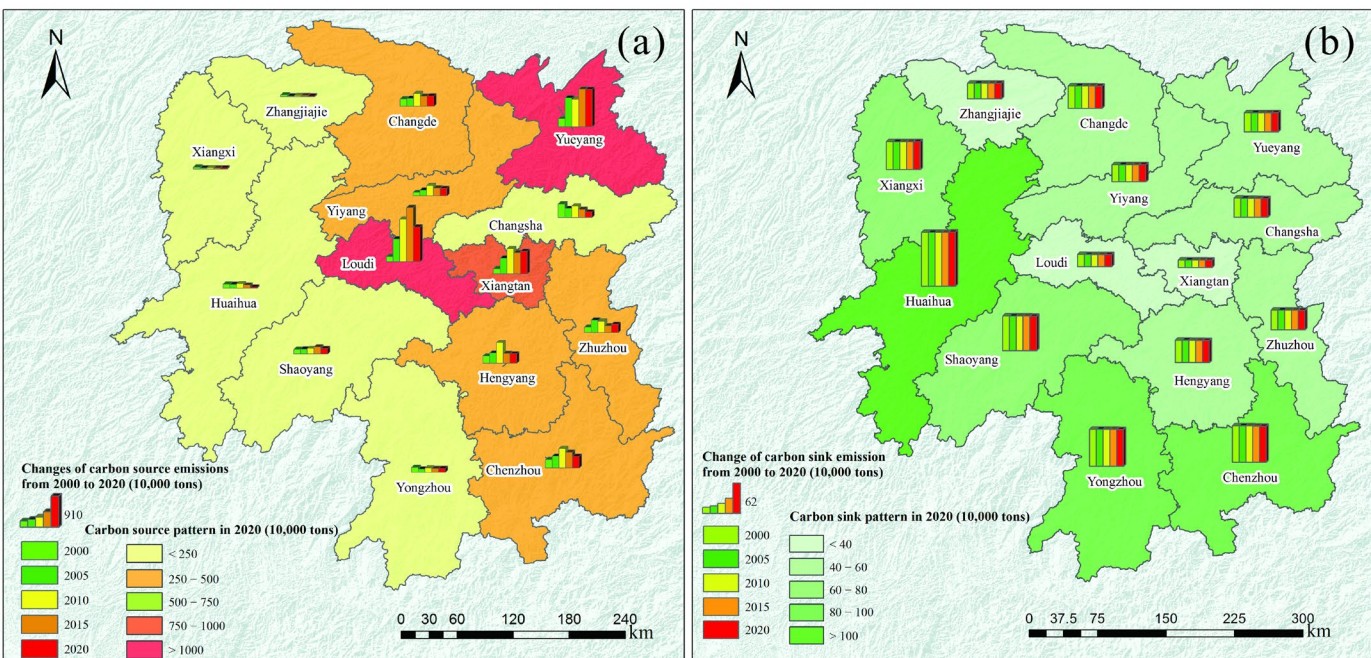

**Figure 7.** (**a**) Spatial model and evolution of carbon sources in Hunan Province from 2000 to 2020; (**b**) spatial model and evolution of carbon sinks in Hunan Province from 2000 to 2020.

### 3.3. Influence of Land Use Change and Socio-Economic Factors on $CO_2$ Equivalent Emissions

In addition to the close relationship between the change in $CO_2$ equivalent emissions and land use change, the relevant research showed that the regional $CO_2$ equivalent emissions are also positively correlated with the social and economic level. Therefore, this study selects the main indicator of social and economic gross national product, and then calculates the net $CO_2$ equivalent emissions per unit GDP, which can more intuitively observe the $CO_2$ equivalent emissions intensity. From the net $CO_2$ equivalent emissions per unit GDP chart (Figure 8), the overall net $CO_2$ equivalent emissions per unit GDP in Hunan Province showed a shrinking and decreasing state from 2000 to 2020. Among them, Loudi's net $CO_2$ equivalent emissions per unit GDP were the highest in Hunan Province during the study period, increasing from $0.87 \times 10^4$ tons/100 million yuan in 2000 to $2.38 \times 10^4$ tons/ 100 million yuan in 2005, and then rapidly decreased to $0.68 \times 10^4$ tons/100 million yuan in 2020, with the most significant decline. Xiangxi, Zhangjiajie, and Huaihua were the cities with the lowest net $CO_2$ equivalent emissions per unit GDP, which had been maintained at the low level of $-0.24 \times 10^4$ tons/100 million yuan to $-0.02 \times 10^4$ tons/100 million yuan for a long time.

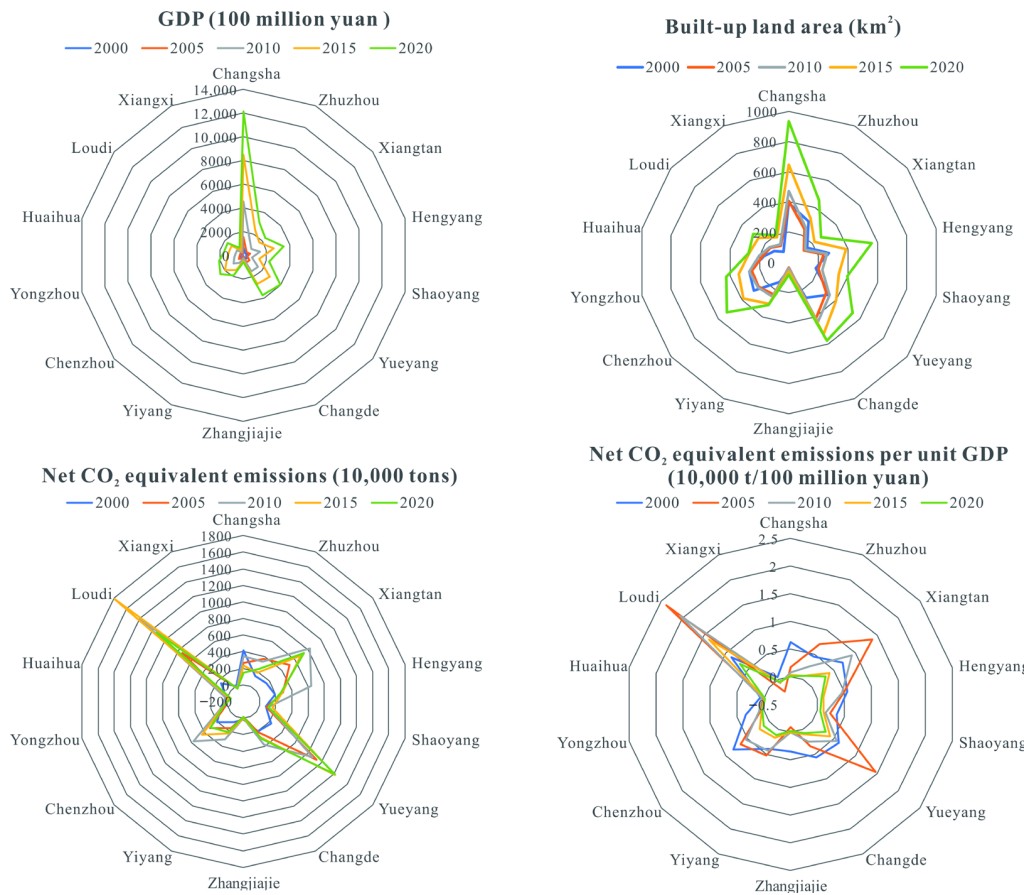

**Figure 8.** Changes in GDP, built-up land area, net CO$_2$ equivalent emissions, and net CO$_2$ equivalent emissions per unit GDP in Hunan Province from 2000 to 2020.

### 3.4. Accounting and Analysis of Carbon Compensation Value

The carbon compensation values of 14 cities (prefectures) in Hunan Province in 2020 are shown in Table 5 and Figure 9. There was considerable similarity between the value of carbon compensation and the spatial distribution of net carbon emissions in each city. The carbon source areas include Yueyang, Loudi, Xiangtan, Chenzhou, Changde, Zhuzhou, Hengyang, Yiyang, Changsha, Shaoyang, and Yongzhou, while the carbon sink areas include Zhangjiajie, Xiangxi, and Huaihua. The carbon source zone is spread throughout the central and eastern parts of Hunan Province, centered around the Changsha–Zhuzhou–Xiangtan economic plate, forming an adjacent uninterrupted large area with its neighboring cities of Hengyang, Yueyang, and Loudi. These carbon source areas, due to their own carbon source exceeding carbon sink, bring carbon "spillovers", and these "spillovers" are unconditionally directly absorbed by adjacent carbon sink areas, so they belong to environmental beneficiaries who need to pay compensation. The carbon sink areas are mainly distributed in the northwest and west of Hunan Province and are located outside the carbon source areas. Carbon sink areas, due to their own carbon sinks being greater than their carbon sources, not only absorb their own carbon emissions, but also absorb the "overflow" of carbon emissions from carbon source areas. Therefore, these cities are ecological protectors and can receive corresponding economic compensation. Among them, Yueyang needed to pay the most carbon compensation funds, followed by Loudi and Xiangtan; they needed to pay 105.2507 million yuan, 76.3758 million yuan, and 39.2963 million yuan respectively, which was directly related to the fact that the energy structure of the three cities is still dominated by fossil energy, the industrial structure is not improved, and the net CO$_2$ equivalent emissions are large. On the contrary, Changsha and Zhuzhou received carbon compensation funds because of their high level of economic

development, a wide range of scientific and technological innovation industries, and energy structure adjustment. According to the different carbon compensation values of different cities in Hunan Province, the cities can be roughly divided into high paid compensation areas, low paid compensation areas, high obtained compensation areas, and low obtained compensation areas. The high paid compensation area included Yueyang, the low paid compensation area included Loudi and Xiangtan, the high obtained compensation area included Changsha, Huaihua, Yongzhou, Hengyang, Shaoyang, Changde, and Zhuzhou, and the low obtained compensation area included Xiangxi, Zhangjiajie, Yiyang, and Chenzhou. From the relationship between carbon emissions and carbon compensation value, the more carbon emissions, the more carbon compensation funds paid. On the contrary, the less carbon emissions, the more carbon compensation funds obtained. From the perspective of regional economic development and carbon compensation value, in order to achieve regional coordination and low-carbon development, it is necessary to continuously narrow the economic gap, to strive for equitable development in the region, and to rely on carbon compensation to promote regional low-carbon coordinated development.

**Table 5.** Carbon compensation value of land use in Hunan Province in 2020.

| City | Changsha | Zhuzhou | Xiangtan | Hengyang | Shaoyang | Yueyang | Changde | Zhangjiajie | Yiyang | Chenzhou | Yongzhou | Huaihua | Loudi | Xiangxi |
|---|---|---|---|---|---|---|---|---|---|---|---|---|---|---|
| carbon compensation funds/$10^4$ yuan | −39,881.78 | −1220.55 | 3929.63 | −1427.29 | −1372.69 | 10,525.07 | −1353.26 | −672.52 | −317.94 | −37.53 | −1849.35 | −2008.47 | 7637.58 | −814.85 |

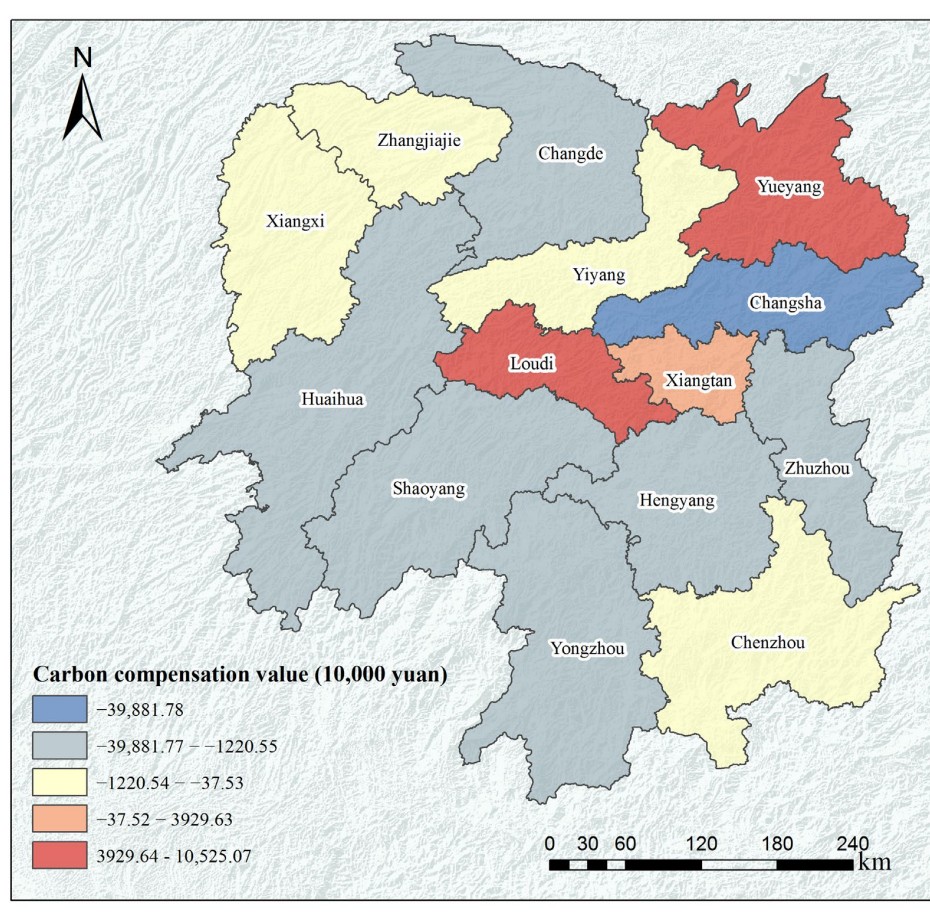

**Figure 9.** Spatial pattern of carbon compensation value of land use in Hunan Province in 2020.

## 4. Discussion and Conclusions

### 4.1. Discussion

The cropland in Hunan Province is mainly concentrated in the Dongting Lake Plain in northern Hunan and Changde, Shaoyang, Yongzhou, Hengyang, and other areas in

the hilly basin areas in central Hunan, while the built-up land is mainly concentrated in Changsha, Zhuzhou, Changde, Yueyang, Hengyang, and other areas of the Changsha–Zhuzhou–Xiangtan urban agglomeration, which is similar to the existing research results at home and abroad [52,53]. The areas with high land use, dense population, high economic development level, and large energy consumption generate large carbon emissions and small forest area and small carbon sinks resulting in low regional carbon compensation rate and weak carbon sink capacity. Forest and grassland are mainly concentrated in Xiangxi, Zhangjiajie, Yongzhou, and Huaihua, which are mostly located in mountainous areas, with low land use level, low urban development level, and a large proportion of forest area, mainly undertaking ecological functions, with large carbon sinks and strong carbon sink capacity.

Carbon compensation is a new field of research related to ecological compensation in the context of global change and the low-carbon economy [54], which is a combination of theoretical conception of ecological compensation and low-carbon action [55]. Based on the concept and connotation of ecological compensation, carbon compensation can be defined as the behavior of carbon emitters to compensate for carbon sinks or ecological protectors economically or uneconomically [56]. The earliest international level of carbon trading compensation can be traced back to the 1990s, when the Kyoto Protocol called for the trading of six greenhouse gas emissions reductions [57]. The objective of carbon compensation is to promote the reduction of carbon emissions and to achieve regional equity and sustainability, which is essentially a low-carbon development model with carbon as the link.

Carbon compensation is usually determined by net $CO_2$ equivalent emissions or net carbon sinks combined with the unit price of carbon sinks, in order to quantify the economic value of the carbon effect of social production and living. Carbon deficit areas should pay a corresponding amount to compensate for the negative impact on the ecological environment, while carbon surplus areas should pay an amount equal to "reward" the contribution of the ecological environment. From the carbon compensation mechanism, the very essence of carbon compensation is to further promote social and ecological equity through the redistribution of resources.

From the regional economic development and carbon compensation value, to achieve regional coordination and low carbon development, it is necessary for Yueyang, Loudi, Xiangtan, and other carbon compensation areas with large net $CO_2$ equivalent emissions and backward industries to pay carbon compensation "funds" to the compensated areas with low economic development level and small net $CO_2$ equivalent emissions, continuously narrow the economic gap, strive to achieve fair development within the region, and rely on carbon compensation to promote regional low-carbon coordinated development [37]. At the same time, land use emissions are affected by carbon sinks and carbon sources. However, the sink capacity of forest areas, as the main carbon sink, is far lower than the carbon emissions of built-up land, so it is impossible to achieve carbon balance. Consequently, there is a need to control fossil fuel usage, to adjust the structure of energy consumption, to develop clean energy, to reduce carbon sources, to increase the area of forest and grassland, and to gradually achieve carbon balance.

In combination with foreign carbon compensation systems and relevant effective practical projects [58,59], in order to achieve regional carbon emission balance and reduction, the following key aspects can be taken into account: (1) Establish and improve regional horizontal carbon compensation systems and mechanisms, determine carbon compensation standards, and set up a special interval for the calculation, collection, distribution, and management of ecological compensation to ensure the functioning of ecological compensation operations; (2) require each region to establish carbon archives, conduct nationwide carbon reduction publicity and education, and enhance regional carbon reserves and carbon compensation awareness; (3) formulate interval ecological compensation policies, laws and regulations, and legally confirm the absolute rights of "carbon emission rights", provide the compensation basis, principles, disciplines, procedures, and implementation details for interregional ecological compensation activities, and provide strong protection for interregional ecological compensation activities; (4) establish an interregional ecological compensation

supervision, consultation, and sanctions agency to supervise the behavior of all parties involved in ecological compensation in a timely manner, and impose timely sanctions on illegal, irregular, and undisciplined behaviors to encourage the normal functioning of ecological compensation activities. Thus, it promotes the coordination of environmental and economic development in various regions, ensures equity in environmental and economic development among regions, and achieves a reduction in regional carbon emissions.

There are obvious differences in carbon emissions among different cities (prefectures) in Hunan Province, showing signs of improvement overall. For a long time, the main energy source in the process of industrialization in Hunan Province relies on coal, which is due to the abundant coal resources and superior price in China. In the future, industry will continue to dominate the industrial structure, and coal will continue to be a major contributor to energy usage in the short term. However, the growth of carbon dioxide can be reduced through energy conservation and improvements in energy use efficiency. Specifically, the government needs to increase investment, control and gradually reduce the proportion of coal in the structure of energy use in accordance with the natural allocation of energy resources in Hunan Province, develop and promote the production technology of green environmental protection, improve the level of clean production, and strengthen the development and use of low-carbon and zero-carbon energy, such as photovoltaic power generation, wind power, nuclear power, biomass energy, hydropower, and geothermal energy, establish a low-carbon energy structure, and ensure coordinated and sustainable economic, resource, and environmental development.

In the future, Hunan's economy will continue to grow at medium and high speeds, and the urbanization rate will be further improved. Built-up land will continue to be the primary source of carbon in Hunan Province and the accumulation area of the three major sectors of industry, construction, and transportation. At the same time, reducing carbon emissions caused by the disorderly expansion of built-up land is also essential. On the one hand, by regulating and optimizing the internal structure and layout of the built-up land, carbon pollution from high carbon emissions is reduced, and priority is given to providing land for emerging industrial projects with low energy consumption, low emissions, and high technology content, and the land for traditional industrial projects with high energy consumption, high pollution, low efficiency, and explicit government elimination has been limited and progressively reduced. On the other hand, there is a need to further increase the areas of forest and grassland, to improve the capacity of carbon sinks, and to promote the rationalization and reduction of carbon emissions in land use planning.

Combining IPCC and related the literatures, the $CO_2$ equivalent emissions coefficient in areas similar to the natural conditions of Hunan Province is selected, and a reasonable accounting system for land use emissions is established. The data results are consistent with the existing research [43,53], and the carbon compensation model is further constructed to divide the carbon compensation value. The results are reasonable. However, Hunan Province, as a province with a high forest vegetation coverage rate, has a different carbon sink capacity of different vegetation, so the calculation of forest $CO_2$ equivalent emissions should be further refined in future research [54,60]. Secondly, when dividing the value of carbon compensation, taking the city as a unit is not conducive to encourage more targeted low-carbon development strategies in various regions. In the future, it should go further on a small county scale, and propose targeted low-carbon development strategies in combination with low-carbon optimization of the urban industrial structure [61].

*4.2. Conclusions*

(1)  Between 2000 and 2020, the net $CO_2$ equivalent emissions from land use in Hunan Province increased and then decreased. Built-up land is the main carbon source, and its $CO_2$ equivalent emissions increase by nearly 2.1 times. Forest land is the main carbon sink, and this carbon sink decreases with the reduction in forest land. The amount of carbon absorbed by carbon sinks is far lower than the carbon sources, which cannot achieve carbon balance and leads to large net $CO_2$ equivalent emissions.

(2)    According to the spatial distribution of net $CO_2$ equivalent emissions in Hunan Province, the areas with high net $CO_2$ equivalent emissions are mainly distributed in the old industrial cities with large $CO_2$ equivalent emissions and small carbon absorption, mainly in Yueyang, Xiangtan, and Loudi, while the areas with low net $CO_2$ equivalent emissions are mainly distributed in Huaihua, Xiangxi, and Zhangjiajie with small $CO_2$ equivalent emissions and widely distributed forest.

(3)    The carbon compensation value is highly similar to the spatial distribution of net $CO_2$ equivalent emissions in each city. According to the different carbon compensation values of each city in Hunan Province, each city in Hunan Province can be divided into a high paid compensation area, low paid compensation area, high obtained compensation area, and low obtained compensation area. From the relationship between carbon emissions and carbon compensation value, the more carbon emissions, the more carbon compensation funds paid. On the contrary, the less carbon emissions, the more carbon compensation funds obtained. Therefore, to achieve regional coordinated development and low-carbon development, it is necessary to establish a government-led regional horizontal carbon compensation system, to continuously narrow the economic gap, strive to achieve fair development within the region, and rely on carbon compensation to promote regional low-carbon oriented development. In parallel, regions with high carbon emissions should also improve energy efficiency, adjust the energy consumption structure, and reduce carbon sources.

**Supplementary Materials:** The following supporting information can be downloaded at: https://www.mdpi.com/article/10.3390/su15107832/s1, Table S1: Land use classification, Table S2: Standard coal conversion coefficient and carbon emission coefficient of various energy sources.

**Author Contributions:** Conceptualization, H.G. and Y.Z.; methodology, H.G., L.H. and Y.L.; software, H.G., H.X. and Z.L.; formal analysis, H.G. and H.X.; investigation, Y.L., H.X. and Z.L.; writing—original draft preparation, H.G., Y.L. and L.H.; writing—review and editing, L.H. and Y.Z.; visualization, H.G., H.X. and Z.L.; supervision, Y.Z. All authors have read and agreed to the published version of the manuscript.

**Funding:** This research was supported by the Social and Science Fund of Hunan Province (Grant No. 18JD26), Hunan Provincial Natural Science Foundation of China (Grant No. 2022JJ50063, 2022JJ50058, 2021JJ50057), and Key Projects of Hunan Provincial Department of Education (Grant No. 22A0419).

**Institutional Review Board Statement:** Not applicable.

**Informed Consent Statement:** Not applicable.

**Data Availability Statement:** The basic data in the research can be found on the website of National Bureau of Statistics, *Hunan Statistical Yearbook*, and the Resource and Environmental Science and Data Center of the Chinese Academy of Sciences (http://www.resdc.cn (accessed on 20 August 2022)).

**Conflicts of Interest:** The authors declare no conflict of interest.

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
