# Peer review of "Temporal and Spatial Differences in CO2 Equivalent Emissions and Carbon Compensation Caused by Land Use Changes and Industrial Development in Hunan Province"

_sustainability, doi:10.3390/su15107832_

Round 1

Reviewer 1 Report

The article attempts to develop the scientific background for regional GHG emissions accounting compatible with national reporting. The authors have the ambition to develop a methodology for multilevel national reporting on the GHG sources and sinks and use it as background for developing economic incentives for GHG emissions reduction. This fact makes this article of great interest and use; however, to make it of this use, the authors are very welcome to make some significant improvements. The recommendations are attached.

Author Response

Dear Reviewer:

Thank you for your letter and for the reviewers’ comments concerning our manuscript entitled “Temporal and spatial differences in CO2 emissions and carbon compensation caused by land use changes and industrial development in Hunan Province” (ID: sustainability-2264659). Those comments are all valuable and very helpful for revising and improving our paper, as well as the important guiding significance to our researches. We have studied comments carefully and have made correction which we hope meet with approval. Revised portion are marked up using the “Track Changes” function in the paper. The main corrections in the paper and the responds to the reviewers’ comments are as following:

Responses to the comments of Reviewer (C, Comments; R, Response):

C: 1. The term “carbon emissions” that the authors use in the article, including the heading of the article, is jargon. We strongly recommend using one of the IPCC glossaries, for example, the glossary for the Sixth Assessment Report, as cited below. That will help keep terminology consistent with the national and business reporting. We know that many articles coming from China use the “carbon emissions” term. When authors would like to keep it, it is strongly recommended that the definition is provided in the “materials and methods”. Please use the publications below to improve your terminology related to climate change problem.

R: Thanks very much to the reviewer for this suggestion. We have modified Terminology according to the IPCC report, such as CO2 emission, cropland, built-up land. See the revised manuscript.

C: 2. Several other terms that appeared possibly due to the direct translation from Chinese and are not coordinated with the international standards:

“Carbon emission coefficient” (t·hm-2) demands an explanation. There is no sin in using hectometres instead of hectares; all is understandable. The question remains about the time dimension. Is the unit related to time? The IPCC guidelines deal with the “emission factors” (TCO2eq*ha-2year-2). When the units are different, please designate why and how they are related to the units used by IPCC in the guidelines for the National Reporting.

R: Thanks very much to the reviewer for this suggestion. We have standardized the unit of “carbon emission coefficient” and added the concept of “carbon emission coefficient” used in the manuscript. See Line 200-206.

C: 3. Swamp – please replace with Wetlands – the land category used by IPCC. “Swamping” in relation to drylands should be replaced by “rewetting” – which is used by IPCC as a contrary activity to “drainage”.

Other land categories, such as farmlands, sometimes appear without definition and designation of relation to the standard IPCC land use categories. Please list all your land categories and show the relations to the IPCC land categories. When they differ from the IPCC ones – please give a clear definition. And further in the text – please use them consistently, do not replace one with the other, or with synonyms.

R: Thanks very much to the reviewer for this suggestion. We have modified Terminology according to the IPCC report. See the revised manuscript.

C: 4. The “Introduction” does not serve its task first because of numerous language confusions and second because the goals and objectives of the current study are not identified clearly here.

The language of the literature review should be checked by a native speaker. The summaries of reviewed publications are not clear and sound contradictory. The reviewed articles should actually support the need of your study. Please summarise your literature review in the way that the need of your study is apparent, or that you are inspired to formulate a new hypothesis, or at least you would like to repeat the proposed by other authors method with some improvement.

The final part of the introduction should be designed so that the hypothesis is formulated and the goals and objectives of the study to support this hypothesis are listed clearly, possibly even as bullet points. The introduction should lead one through the entire study in a way that you could connect the statements in the introduction with the other parts of the article – such as methods and results. Please identify clear objectives and use those objectives in the further text as headings or subheadings of your text parts.

R: Thanks very much to the reviewer for this suggestion. We have reorganized the introduction section. See the Line 40-122.

C: 5. Please remember that most of the readers are not familiar with Chinese reality and geography Would be nice to place the schematic map of the entire China and designate the position of the pilot regions on the countries map.

R: Thanks very much to the reviewer for this suggestion. We have added the schematic map of the entire China and Hunan Province. See Figure 1.

C: 6. Naming your areas under study “cities” misleads the reader, as one expects that authors are dealing with the cities; however, in the study are included lands of the entire provinces or prefectures. Please clarify this in the materials and methods.

R: Thanks very much to the reviewer for this suggestion. We have added detailed description of the study object. See Line 125-149.

C: 7. Under the heading “Data sources”, the description of land classes is presented. It is better to introduce a separate part for land classification methodology. Please describe the method of how the land classes had been identified, list the land classes and give clear definitions. In the current version of the article, authors refer to two levels of classes; one level has six classes, and the other level – has 24 classes. Would be helpful to explain how authors came to this system.

R: Thanks very much to the reviewer for this suggestion. The land use classification is listed in the Supplementary Material. See Supplementary Material and Line 161-163.

C: 8. Part 2.3 strongly demands language improvement, including the main heading and subheadings. Most of the text is confusing because of the formulation difficulties, mainly because of the word order in the phrases. For example, “2.3.1 Dynamic degree analysis of land use change” – is not understandable. Only when one reads the chapter – it becomes clear that the method how to assess the land use change rate is described.

The other example of total confusion – is the heading “calculation method of direct carbon emission” and “calculation method of indirect carbon emission”. The author does not mean direct and indirect emissions. But direct and indirect methods, correct? However, the heading states differently.

R: Thanks very much to the reviewer for this suggestion. We have modified the description of Part 2.3. See Line 173-283.

C: 9. Results. Please check language and remove all elements of the discussion.

R: Thanks very much to the reviewer for this suggestion. We have modified the description and reorganized the Results section. See Line 284-555.

C: 10. Discussion and conclusion. The first part summarises results – that is, repetition and misplaced part. Please concentrate on discussion and recommendations.

R: Thanks very much to the reviewer for this suggestion. We have reorganized the Discussion and Conclusion section. See Line 556-694.

C: 11. Comments on the suggested methodology

R: Thanks very much to the reviewer for this suggestion. We have reorganized the Discussion and Conclusion section. See Line 556-694.

C: 12. Table 1. Would be wise to present the construction lands.

R: Thanks very much to the reviewer for this suggestion. The CO2 emissions from built-up land in this manuscript are calculated by energy consumption, not by the CO2 emission coefficient method similar to that for cropland and forest, and the parameters for calculating CO2 emissions from energy consumption are listed in the Supplementary Material. See Supplementary Material and Line 161-163.

C: 13. Table 2. Language issue – “dynamic degree” change to “rate”.

R: Thanks very much to the reviewer for this suggestion. We have modified it. See Table 2.

C: 14. Table 3 is totally not readable and useless – strongly recommended to change into a graph.

R: Thanks very much to the reviewer for this suggestion. We have changed the Table 3 into a graph. See Figure 2.

C: 15. Table 4. In the heading remove “analysis”. It is not an analysis, just a matrix. Two values after the comma in percentages – is too much.

R: Thanks very much to the reviewer for this suggestion. We have modified it. See Table 3.

C: 16. Table 5 – is not readable. Please reflect the content of the columns in general headings. Please note that there is no sin in calling negative emissions still emissions. It is no need to confuse everybody by changing the term “emission” to “absorption” depending on the “minus” or “plus” sign. The other story is about sinks and sources. What the authors present in Table 5 –are emissions based on the areas of sinks and sources in the understanding of the IPCC national inventory Guidelines.

R: Thanks very much to the reviewer for this suggestion. We have modified it. See Table 4.

C: 17. Figure 1 -as suggested before – please show the position of the province in the entire map of China

R: Thanks very much to the reviewer for this suggestion. We have added the schematic map of the entire China and Hunan Province. See Figure 1.

C: 18. Figure 2 and Figure 3 – there is the question of whether the dynamic was always one way. If the land was deforested, cant it be reforested within those 20 years? the dynamic between grasslands and croplands is even faster. Is it taken into account in the schemes?

R: Thanks very much to the reviewer for this suggestion. We have modified the figure. See Figure 2 and Figure 3.

C: 19. Figure 5 – is too crowded. When you have values on the top of the bars, there is no need in Y-axes with the confusing scale with four zeroes. You can have a heading – annual emissions, and in the legend, you can leave only land class names without mentioning absorption or emission. That is why you are making a graph – it illustrates itself.

R: Thanks very much to the reviewer for this suggestion. We have modified the figure. See Figure 5.

C: 20. The language of articles is not designated. A large number of articles in the reference list are in Chinese. However, it is not designated in the list and misleads one in the search.

R: Thanks very much to the reviewer for this suggestion. We have reorganized the reference. See Reference.

C: 21. Please include references to the IPCC guidelines and look at the literature on land use classification and land use change mapping.

R: Thanks very much to the reviewer for this suggestion. We have reorganized the figures. See Figure 3 and Figure 4.

Reviewer 2 Report

My recommendation for the manuscript is  Major Revision. 

1) The authors should improve the introduction section. The authors should provide more references. 

2) What is UNUSED LAND mean? Is it bare land or? In this study, there are 6 LULC categories. I think giving information about the LULC categories will help understand the study area. For example, what type of forests are there in the region? 

3) What is the accuracy of the LULC data ? The authors should provide information about the data. 

4) The authors should provide a graph to show the LULC changes for the selected years. 

5) Figures 2, 3, and 4 are not understandable. The authors should improve these three important maps. It is not clear how forests, croplands, and artificial surfaces were changed in the study area. 

6) In figure 5 b there are only three categories but legend has five. Please check it. 

7) The authors should provide a detailed analysis such as spatial analysis to show LULC changes, especially for the Loudi and Yueyang (they have the highest carbon compensation values). 

8) Please check also the journal name it says LAND in the manuscript. 

Author Response

Dear Reviewer:

Thank you for your letter and for the reviewers’ comments concerning our manuscript entitled “Temporal and spatial differences in CO2 emissions and carbon compensation caused by land use changes and industrial development in Hunan Province” (ID: sustainability-2264659). Those comments are all valuable and very helpful for revising and improving our paper, as well as the important guiding significance to our researches. We have studied comments carefully and have made correction which we hope meet with approval. Revised portion are marked up using the “Track Changes” function in the paper. The main corrections in the paper and the responds to the reviewers’ comments are as following:

Responses to the comments of Reviewer (C, Comments; R, Response):

C: 1. The authors should improve the introduction section. The authors should provide more references.

R: Thanks very much to the reviewer for this suggestion. We have reorganized the introduction section. See the Line 40-122.

C: 2. What is UNUSED LAND mean? Is it bare land or? In this study, there are 6 LULC categories. I think giving information about the LULC categories will help understand the study area. For example, what type of forests are there in the region?

R: Thanks very much to the reviewer for this suggestion. The land use classification is listed in the Supplementary Material. See Supplementary Material and Line 154-172.

C: 3. What is the accuracy of the LULC data? The authors should provide information about the data.

R: Thanks very much to the reviewer for this suggestion. The land use data come from the Resource and Environ-mental Science and Data Center of the Chinese Academy of Sciences with a resolution of 30 m × 30 m, produced by manual visual interpretation and supervised classification methods with an accuracy verification of 85.72% and a Kappa coefficient of 0.82. See Line 154-172.

C: 4. The authors should provide a graph to show the LULC changes for the selected years.

R: Thanks very much to the reviewer for this suggestion. We have reorganized the figures. See Figure 4.

C: 5. Figures 2, 3, and 4 are not understandable. The authors should improve these three important maps. It is not clear how forests, croplands, and artificial surfaces were changed in the study area.

R: Thanks very much to the reviewer for this suggestion. We have reorganized the figures. See Figure 2, Figure 3 and Figure 4.

C: 6. In figure 5b there are only three categories but legend has five. Please check it.

R: Thanks very much to the reviewer for this suggestion. We have modified the figure. See Figure 5.

C: 7. The authors should provide a detailed analysis such as spatial analysis to show LULC changes, especially for the Loudi and Yueyang (they have the highest carbon compensation values).

R: Thanks very much to the reviewer for this suggestion. We have added the description of LULC changes. See Line 442-464.

C: 8. Please check also the journal name it says LAND in the manuscript.

R: Because when we were preparing to submit two papers, the office software made a mistake and confused the downloaded submission template, resulting in the word 'Land' appearing in the manuscript. We are very sorry for the inconvenience caused to you. The error has been corrected.

Reviewer 3 Report

Dear Authors, I indicated some minor corrections in the text, you can find my comments in the attached pdf file.

Author Response

Dear Reviewer:

Thank you for your letter and for the reviewers’ comments concerning our manuscript entitled “Temporal and spatial differences in CO2 emissions and carbon compensation caused by land use changes and industrial development in Hunan Province” (ID: sustainability-2264659). Those comments are all valuable and very helpful for revising and improving our paper, as well as the important guiding significance to our researches. We have studied comments carefully and have made correction which we hope meet with approval. Revised portion are marked up using the “Track Changes” function in the paper. The main corrections in the paper and the responds to the reviewers’ comments are as following:

Responses to the comments of Reviewer (C, Comments; R, Response):

C: 1. I indicated some minor corrections in the text, you can find my comments in the attached pdf file.

R: Thanks very much to the reviewer for this suggestion. We have already modified the manuscript according to your comments. See the revised manuscript.

Round 2

Reviewer 1 Report

Dear Authors, thank you for meeting most of my concerns. The logic of the article had significantly improved. However, I still have two suggestions.

1) The first is to check the text once more and appropriately use the names of three entities: carbon, greenhouse gases (GHG) and CO2 equivalent (not CO2). The authors automatically replaced "carbon" with CO2 throughout the entire article text. That needs additional attention. In some cases, you really speak about carbon balance, and then you should use "carbon"; in other contexts - you write about GHG fluxes, and when it goes to fluxes measuring and reporting - only then the "CO2 equivalent" comes to the scene. Please note - not "CO2" but "CO2 equivalent", and only in the cases of measurable or estimated fluxes. It is a unit to reflect the scale of GHG fluxes. Please make appropriate replacements keeping this in mind. Please look again at the concepts behind the terms in the glossary from the latest IPCC report. The current use of CO2 all over the text is incorrect and scientifically not sound.

2) when the National Inventory Reporting of China is mentioned, the authors refer to the Guidelines for National Greenhouse Gas Emission Inventory issued by IPCC in 2006. However, the guidelines are not on the list of references. Please include references to both Guidelines as a full citation - IPCC 2006 and Wetlands Supplement 2013. 

I hope these improvements can be easily met and the article text can be proven by the native speaker and published soon.

Author Response

Responses to the comments of Reviewer #1 (C, Comments; R, Response):

C: 1. The first is to check the text once more and appropriately use the names of three entities: carbon, greenhouse gases (GHG) and CO2 equivalent (not CO2). The authors automatically replaced "carbon" with CO2 throughout the entire article text. That needs additional attention. In some cases, you really speak about carbon balance, and then you should use "carbon"; in other contexts - you write about GHG fluxes, and when it goes to fluxes measuring and reporting - only then the "CO2 equivalent" comes to the scene. Please note - not "CO2" but "CO2 equivalent", and only in the cases of measurable or estimated fluxes. It is a unit to reflect the scale of GHG fluxes. Please make appropriate replacements keeping this in mind. Please look again at the concepts behind the terms in the glossary from the latest IPCC report. The current use of CO2 all over the text is incorrect and scientifically not sound.

R: Thanks very much to the reviewer for this suggestion. We have modified Terminology about “CO2 equivalent emissions”, “carbon balance” and “carbon compensation” according to the IPCC report. See the revised manuscript.

C: 2. when the National Inventory Reporting of China is mentioned, the authors refer to the Guidelines for National Greenhouse Gas Emission Inventory issued by IPCC in 2006. However, the guidelines are not on the list of references. Please include references to both Guidelines as a full citation - IPCC 2006 and Wetlands Supplement 2013.

R: Thanks very much to the reviewer for this suggestion. We have Added corresponding references. See Line 211, 225-226, Reference 42-43.

Reviewer 2 Report

I carefully checked the authors' responses for my comments. My overall recommendation for the revised manuscript is ACCEPT. 

Author Response

Responses to the comments of Reviewer #2 (C, Comments; R, Response):

C: 1. I carefully checked the authors' responses for my comments. My overall recommendation for the revised manuscript is ACCEPT.

R: Thanks very much to the reviewers for their comments and assistance with this manuscript.